# Cryptic susceptibility to penicillin/β-lactamase inhibitor combinations in emerging multidrug-resistant, hospital-adapted *Staphylococcus epidermidis* lineages

Xiaoliang Ba [1,10], Claire L. Raisen[1,10], Olivier Restif [1], Lina Maria Cavaco [2], Carina Vingsbo Lundberg[2], Jean Y. H. Lee [3], Benjamin P. Howden [3], Mette D. Bartels [4,5], Birgit Strommenger[6], Ewan M. Harrison[7,8,9], Anders Rhod Larsen [2], Mark A. Holmes [1,11] & Jesper Larsen [2,11] ✉

Global spread of multidrug-resistant, hospital-adapted *Staphylococcus epidermidis* lineages underscores the need for new therapeutic strategies. Here we show that many *S. epidermidis* isolates belonging to these lineages display cryptic susceptibility to penicillin/β-lactamase inhibitor combinations under in vitro conditions, despite carrying the methicillin resistance gene *mecA*. Using a mouse thigh model of *S. epidermidis* infection, we demonstrate that single-dose treatment with amoxicillin/clavulanic acid significantly reduces methicillin-resistant *S. epidermidis* loads without leading to detectable resistance development. On the other hand, we also show that methicillin-resistant *S. epidermidis* is capable of developing increased resistance to amoxicillin/clavulanic acid during long-term in vitro exposure to these drugs. These findings suggest that penicillin/β-lactamase inhibitor combinations could be a promising therapeutic candidate for treatment of a high proportion of methicillin-resistant *S. epidermidis* infections, although the in vivo risk of resistance development needs to be further addressed before they can be incorporated into clinical trials.

*Staphylococcus epidermidis* is part of the normal human skin microbiota but is also an increasing cause of difficult-to-treat invasive infections around the world due to global spread of multidrug-resistant, hospital-adapted isolates[1–4]. Three *S. epidermidis* sequence types (STs) have become particularly successful in hospital settings, namely ST2 (including ST188), ST5 (including ST87), and ST23[2–4], of which ST2 comprises two sublineages referred to as the BPH0662 clone and ST2-mixed[4]. A large proportion of ST2, ST5, and ST23 isolates is resistant to methicillin and most other β-lactam antibiotics, including penicillinase-labile penicillins (e.g., penicillin G and amoxicillin),

[1]Department of Veterinary Medicine, University of Cambridge, Cambridge, UK. [2]Department of Bacteria, Parasites & Fungi, Statens Serum Institut, Copenhagen, Denmark. [3]Department of Microbiology and Immunology, The University of Melbourne at The Doherty Institute for Infection and Immunity, Melbourne, VIC, Australia. [4]Department of Clinical Microbiology, Copenhagen University Hospital - Amager and Hvidovre, Hvidovre, Denmark. [5]Department of Clinical Medicine, University of Copenhagen, Copenhagen, Denmark. [6]National Reference Centre for Staphylococci and Enterococci, Division Nosocomial Pathogens and Antibiotic Resistances, Department of Infectious Diseases, Robert Koch Institute, Wernigerode Branch, Wernigerode, Germany. [7]Department of Medicine, University of Cambridge, Cambridge, UK. [8]Department of Public Health and Primary Care, University of Cambridge, Cambridge, UK. [9]Wellcome Sanger Institute, Hinxton, UK. [10]These authors contributed equally: Xiaoliang Ba, Claire L. Raisen. [11]These authors jointly supervised this work: Mark A. Holmes, Jesper Larsen. ✉e-mail: jrl@ssi.dk

penicillinase-stable penicillins (e.g., methicillin), and cephalosporins (e.g., cefoxitin), due to carriage of the *mecA* gene on a chromosomally integrated mobile genetic element known as staphylococcal cassette chromosome *mec* (SCC*mec*)[4]. In addition, these methicillin-resistant *S. epidermidis* (MRSE) lineages frequently display reduced susceptibility to vancomycin (the most common drug of choice for treatment of MRSE infections), teicoplanin, and rifampicin (used in combination with other drugs for treatment of staphylococcal medical device-related infections) as well as to macrolides, quinolones, aminoglycosides, and sulfonamides[4]. The increasing prevalence of nearly pan-resistant *S. epidermidis* isolates in at-risk patients is concerning and emphasises the need for new therapeutic options[5].

A recent study showed that several lineages of methicillin-resistant *Staphylococcus aureus* (MRSA) display cryptic susceptibility to penicillins, in particular benzyl- and aminopenicillins (e.g., penicillin G and amoxicillin, respectively), when used in combination with a β-lactamase inhibitor such as clavulanic acid[6]. Genomic analysis revealed that phenotypic susceptibility is caused by unique mutations in the *mecA* promoter region that lower expression of penicillin-binding protein 2a (PBP2a) and/or amino acid substitutions in PBP2a that increase its affinity for penicillins in the presence of clavulanic acid[6]. Overall, that study identified four susceptible (S1–S4) and two resistant (R1 and R2) genotypes in a collection of 384 MRSA isolates from a diverse range of clinically relevant lineages[6]. It is generally accepted that SCC*mec* elements were transferred from *S. epidermidis* to *S. aureus*, although the vehicles and modes of interspecies exchange of DNA remain unclear[7,8]. These findings led us to speculate whether the MRSE ST2, ST5, and ST23 lineages harbour the same *mecA* promoter mutations and PBP2a substitutions and, if so, whether they are susceptible to penicillin/β-lactamase inhibitor combinations in vitro and in vivo.

In this work, we show that many MRSE ST2, ST5, and ST23 isolates have either the S2 or S3 genotype and are phenotypically susceptible to amoxicillin/clavulanic acid under in vitro conditions. We further demonstrate that single-dose administration of amoxicillin/clavulanic acid decreases MRSE counts in a mouse infection model without leading to detectable resistance development. On the other hand, we show that in vitro selection leads to rapid development of resistance to amoxicillin/clavulanic acid. These findings indicate that penicillin/β-lactamase inhibitor combinations hold promise as a potential therapeutic option against MRSE infections but also underscore the need for additional studies to assess the in vivo risk of resistance development.

## Results and discussion
### *S. epidermidis* isolates and genomic analysis
We analysed the genomes from a previously described global collection of 227 *S. epidermidis* isolates from 96 hospitals in 24 countries, including 73 BPH0662 clone, 60 ST2-mixed, 15 ST5 and 50 ST23 isolates[4]. The analysis showed that most BPH0662 clone (73/73), ST2-mixed (59/60), ST5 (15/15), and ST23 (47/50) isolates carried *mecA* (Supplementary Data 1). All 62 MRSE ST5 and ST23 isolates had the S2 genotype, while 26 of the 59 MRSE ST2-mixed isolates had either the S2 (*n* = 21) or S3 (*n* = 5) genotype (hereafter referred to as S2/S3 isolates). The remaining 33 MRSE ST2-mixed and all 73 MRSE BPH0662 clone isolates had the R2 genotype (hereafter referred to as R2 isolates), except for one MRSE ST2-mixed isolate that could not be genotyped due to the presence of the insertion sequence IS*256* at position −15 in the *mecA* promoter. Of the 29 isolates belonging to other lineages, 16 had either the S2 (*n* = 15) or S3 (*n* = 1) genotype, whereas five had the R2 genotype. The eight remaining isolates lacked *mecA*. The *blaZ* gene encoding β-lactamase enzyme was present in 212 of the 227 isolates, of which seven carried two copies, while it was absent in 14 isolates and truncated in one isolate due to a frameshift deletion at position 100 leading to a premature stop codon (TAA) at position 103–105. Our findings that *S. epidermidis* and *S. aureus* harbour the same *mecA*

promoter mutations and PBP2a substitutions are consistent with the generally accepted view that SCC*mec* elements in *S. aureus* originated from *S. epidermidis*[7,8].

### Cryptic susceptibility to amoxicillin/clavulanic acid
All MRSE BPH0662 clone, ST2-mixed, ST5, and ST23 isolates from Australia, Denmark, and Germany (*n* = 138) were selected for further analysis (Supplementary Data 2). In the original work on MRSA, amoxicillin/clavulanic acid minimum inhibitory concentrations (MICs) in *S. aureus* were determined using amoxicillin Etest strips applied to Iso-Sensitest agar (ISA) plates supplemented with 15 µg ml⁻¹ clavulanic acid[6]. To our surprise, we observed poor growth of several *S. epidermidis* isolates on these plates, which prevented us from determining MICs with this method, and we therefore switched to MIC determination by microdilution in Iso-Sensitest broth (ISB) with or without 15 µg ml⁻¹ clavulanic acid. We first sought to determine the penicillin G and penicillin G/clavulanic acid MICs in a subset of 43 MRSE isolates, which showed that the presence of 15 µg ml⁻¹ clavulanic acid reduced the MICs below the Clinical and Laboratory Standards Institute breakpoint for penicillin G (≤0.125 µg ml⁻¹)[9] in 55% (11/20) of the S2/S3 isolates but in only 4.3% (1/23) of the R2 isolates (Fig. 1a, b). Intravenous treatment with Augmentin, which contains amoxicillin/clavulanic and is also known as co-amoxiclav, has been shown to significantly reduce the bacterial load in mice infected with an MRSA strain USA300 that had the S2 genotype and was phenotypically susceptible to penicillin/clavulanic acid[6]. MIC determinations of all 138 MRSE isolates showed that the amoxicillin MICs were generally decreased in the presence of 15 µg ml⁻¹ clavulanic acid (Fig. 2a, b), which is consistent with the trend that was observed for penicillin G (Fig. 1a, b). There was a strong correlation between the MICs of amoxicillin/clavulanic acid and other β-lactam antibiotics, including amoxicillin, clavulanic acid, penicillin G, and cefoxitin (Fig. 3).

The breakpoint for amoxicillin/clavulanic acid has not been established for *S. epidermidis*, and we therefore determined an empirical epidemiological cut-off (ECOFF) value using the MICs for all 138 MRSE isolates (Fig. 2a, b). In the presence of 15 µg ml⁻¹ clavulanic acid, 71 (51%) of the 138 MRSE isolates had amoxicillin MICs at or below the empirical ECOFF of 4 µg ml⁻¹ (rounded down from 5.66 µg ml⁻¹) and were consequently deemed susceptible, including 10% (6/58) of the BPH0662 clone isolates, 79% (33/42) of the ST2-mixed isolates, 83% (5/6) of the ST5 isolates, and 84% (27/32) of the ST23 isolates (Table 1). The use of the three genotypes (R2, S2, and S3) to predict susceptibility was accurate in 83% (114/138) of the isolates (Table 1), with a 5.1% (7/138) very major error rate (defined as isolates that were phenotypically resistant, but genotypically predicted to be susceptible) and a 12% (17/138) major error rate (phenotypically susceptible, but genotypically predicted resistant).

### In vivo activity of amoxicillin/clavulanic acid
The empirical ECOFF for amoxicillin in the presence of 15 µg ml⁻¹ clavulanic acid (4.0 µg ml⁻¹) lies in the "susceptible, increased exposure" category according to the European Committee on Antimicrobial Susceptibility Testing (EUCAST) pharmacokinetic-pharmacodynamic (PK-PD) breakpoints for amoxicillin alone as well as for amoxicillin in the presence of 2 µg ml⁻¹ clavulanic acid (susceptible, ≤2 µg ml⁻¹; resistant, >8 µg ml⁻¹). This suggests a high likelihood of therapeutic success but only when a higher than normal dosage can be used or when the agent is physiologically concentrated at the site of infection[10,11]. We explored this possibility in a mouse thigh model of *S. epidermidis* infection using DEN09, an S2 isolate belonging to ST23 and having a MIC of 4 µg ml⁻¹. Treatment with a single dose of 100 mg kg⁻¹ or 250 mg kg⁻¹ amoxicillin did not reduce the bacterial loads compared with vehicle treatment (Fig. 2c). In contrast, treatment with a single dose of 250 mg kg⁻¹ amoxicillin in combination with 50 mg kg⁻¹ clavulanic acid significantly reduced the bacterial loads to a level similar

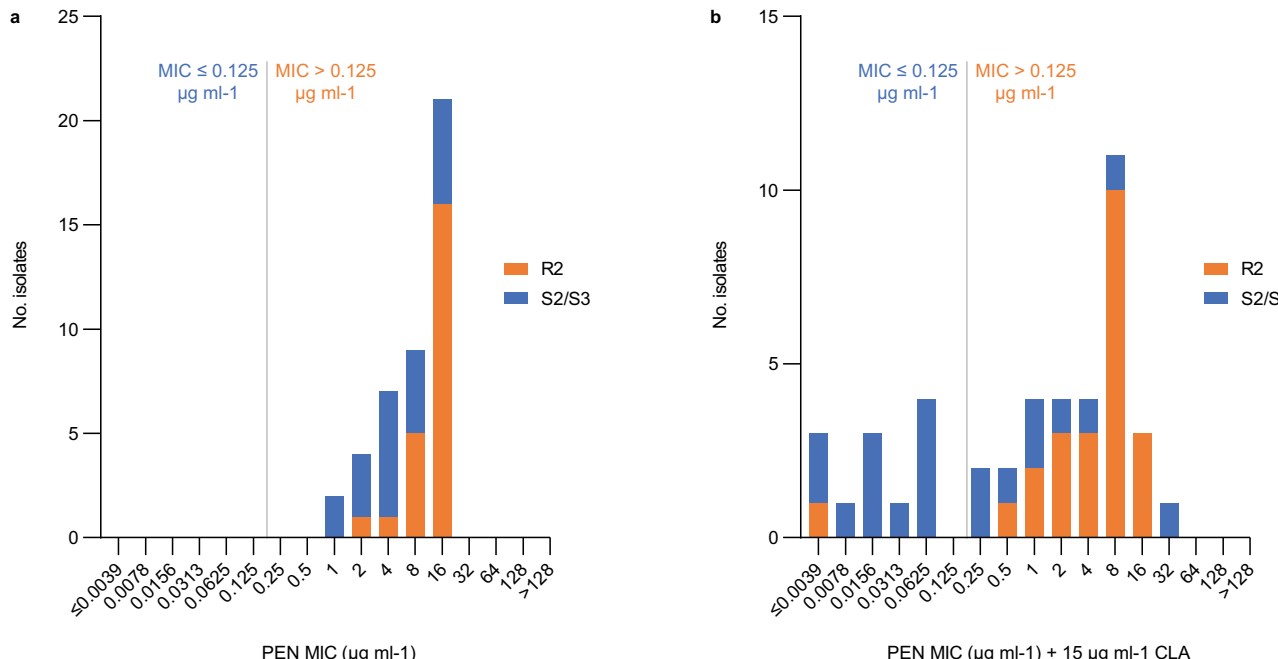

**Fig. 1 | Penicillin G susceptibillin in the presence of 15 μg ml⁻¹ clavulanic acid.**
**a** Broth microdilution determination of the minimum inhibitory concentration of penicillin G alone in a subset of 43 methicillin-resistant *Staphylococcus epidermidis* isolates, including 20 S2/S3 isolates and 23 R2 isolates. **b** Effect of clavulanic acid on susceptibility to penicillin. The Clinical and Laboratory Standards Institute break-point for penicillin G (≤0.125 μg ml⁻¹) is shown as grey vertical lines (**a**, **b**). PEN penicillin G, CLA clavulanic acid, MIC minimum inhibitory concentration. Source data are provided as a Source Data file.

to treatment with 40 mg kg⁻¹ vancomycin ($P = 0.0003$, Dunnett's multiple comparisons test) (Fig. 2c).

Genotypically susceptible MRSA isolates are known to display heteroresistance to penicillin G/clavulanic acid, meaning that they contain subpopulations of mutants with substantially reduced antibiotic susceptibility compared with the main wild-type cell populations[6]. It has previously been shown that resistance to some synergistic drug combinations evolves faster than resistance to the individual drugs alone due to a larger selective advantage for resistant mutants in competition with susceptible wild-type cells[12]. Population analysis profile (PAP) testing of DEN09 isolates recovered from mice treated with a single dose of vehicle alone or 250 mg kg⁻¹ amoxicillin in combination with 50 mg kg⁻¹ clavulanic acid supported the existence of mixed populations in both treatment groups (Fig. 2d). The fraction of resistant cells was similar in both populations based on the PAP-area under the curve (AUC) method ($AUC_{vehicle} = 30.05$; $AUC_{amoxicillin/clavulanic\ acid} = 29.96$; $P = 0.8958$, two-tailed unpaired Student's *t* test), suggesting that single-dose treatment with amoxicillin/clavulanic acid does not select for resistant mutants.

### Development of resistance to amoxicillin/clavulanic acid
Incomplete congruence between *mecA* promoter mutations and PBP2a substitutions and amoxicillin/clavulanic acid susceptibility led us to search for alterations in the β-lactamase enzyme (the target of clavulanic acid) and native PBPs (the targets of all β-lactam antibiotics) in the seven isolates that were phenotypically resistant, but genotypically predicted to be susceptible (Supplementary Table 1). All the isolates had the S2 genotype and belonged to ST2-mixed ($n = 1$), ST5 ($n = 1$), and ST23 ($n = 5$). A maximum-likelihood phylogeny revealed that the five ST23 isolates, DEN22, DEN30, DEN31, DEN35, and GER08, formed a separate cluster (Fig. 4). Comparison with their closest phenotypically susceptible neighbour on the phylogenetic tree, AUS27, showed that all five ST23 isolates had lost *blaZ* (Fig. 4). However, DEN24, an unrelated phenotypically susceptible S2 isolate belonging to ST23, also

lacked *blaZ* (Fig. 4), and it is therefore unclear whether and how the presence/absence of *blaZ* might affect susceptibility to penicillin/β-lactamase inhibitor combinations. In addition, all the phenotypically resistant, but genotypically susceptible ST23 isolates had an S699L substitution in PBP2 when compared with AUS27, while GER08 had an additional D718V substitution in PBP1. The ST2-mixed isolate, AUS16, had a Q145K substitution in PBP3, whereas we did not find any substitutions in the native PBPs of the ST5 isolate, BPH0723. These results raise the possibility that MRSE is capable of developing resistance to amoxicillin/clavulanic acid over longer periods of antibiotic exposure, although it should be noted that the substitutions were located outside the conserved penicillin-binding motifs in the transpeptidase domain of the native PBPs.

We further investigated this possibility by serially passaging BPH0719 and DEN09 in 15 μg ml⁻¹ clavulanic acid with increasing concentrations of amoxicillin over 30 days to select for spontaneous resistant mutants. BPH0719, an S2 isolate belonging to ST2-mixed and having a 256-fold lower MIC than DEN09, initially displayed rapid resistance development, which was followed by a steadier increase (Fig. 5a). DEN09 displayed a slow increase in resistance (Fig. 5b). To investigate whether the ability to develop resistance to amoxicillin/clavulanic acid is a common feature of MRSE under in vitro conditions, we determined the initial MIC and the mutant prevention concentration (MPC) of three other S2 isolates, AUS27 (ST23; MIC = 8 μg ml⁻¹), BPH0724 (ST2-mixed; MIC = 0.125 μg ml⁻¹), and GER21 (ST5; MIC = 1 μg ml⁻¹). All three isolates had an MPC value of 16 μg ml⁻¹, which is consistent with the results of our serial passaging experiments. DEN09 displays heteroresistance to vancomycin and is furthermore resistant to rifampicin due to dual D471E and I527M substitutions in RpoB, whereas BPH0719 is fully susceptible to these drugs[4]. Serial passaging in the presence of these drugs showed that both BPH0719 and DEN09 developed increased resistance to vancomycin over 30 days (Fig. 5a, b), and that BPH0719 frequently developed resistance to rifampicin within a few days of exposure (Fig. 5a).

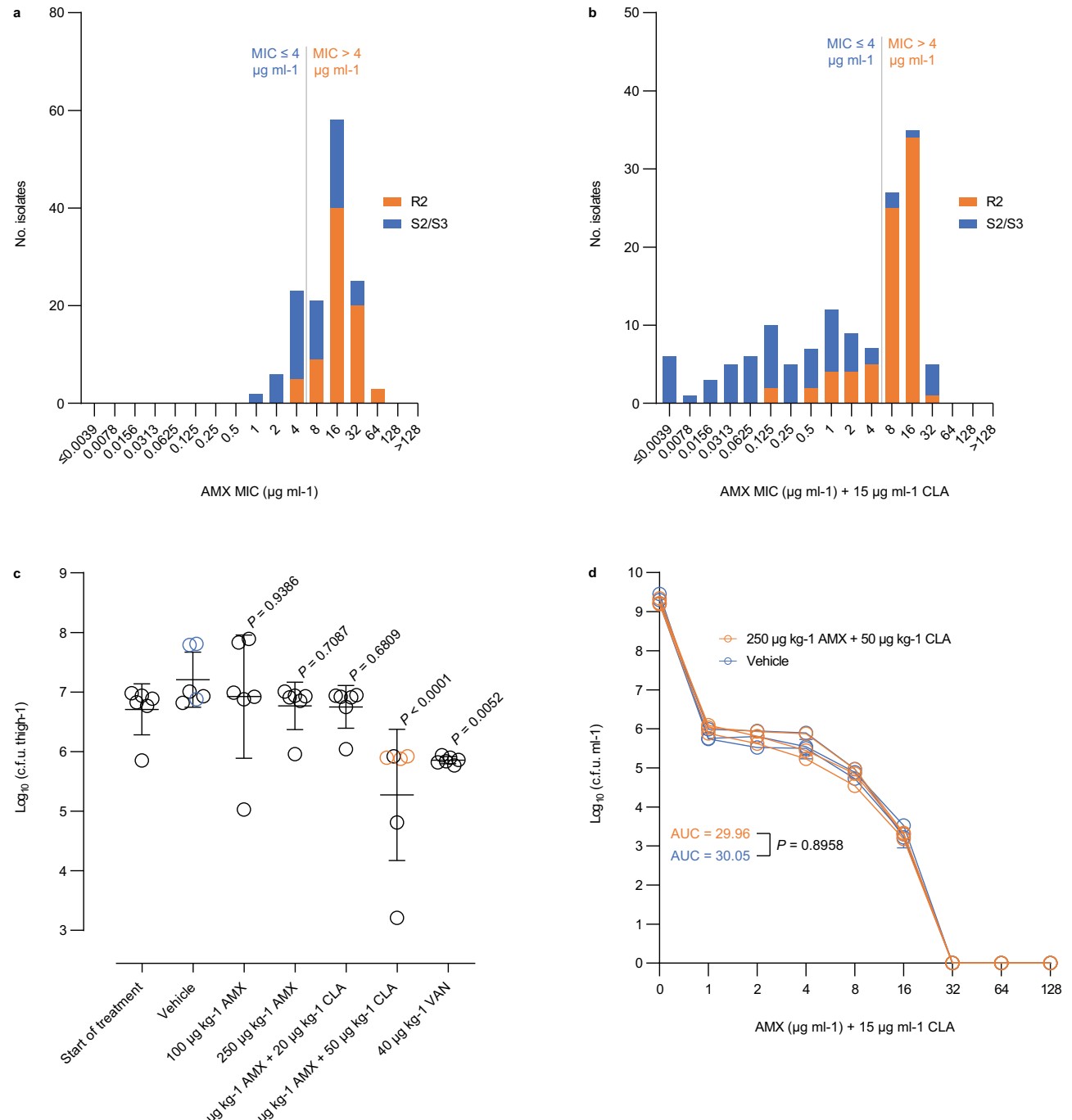

**Fig. 2 | Amoxicillin susceptibility in the presence of 15 µg ml⁻¹ clavulanic acid.**
**a** Broth microdilution determination of the minimum inhibitory concentration of amoxicillin alone in the 138 methicillin-resistant *Staphylococcus epidermidis* isolates, including 61 S2/S3 isolates and 77 R2 isolates. **b** Effect of clavulanic acid on susceptibility to amoxicillin. The empirical epidemiological cut-off for amoxicillin in the presence of 15 µg ml⁻¹ clavulanic acid (4.0 µg ml⁻¹) is shown as grey vertical lines (**a**, **b**). **c** Effect of amoxicillin alone or in combination with clavulanic acid against DEN09 in a mouse infection model. Mice were inoculated intramuscularly with around 8.5 × 10⁷ colony-forming units and treated subcutaneously with a single dose of the indicated drugs. Dunnett's multiple comparisons tests were used to

compare the effect of amoxicillin alone or in combination with clavulanic acid against vehicle treatment. Data are mean ± s.d. with *n* = 6 mice per group. **d** Population analysis profiles (PAPs) of DEN09 isolates recovered from mice treated with a single dose of vehicle alone (*n* = 3) or 250 mg kg⁻¹ amoxicillin in combination with 50 mg kg⁻¹ clavulanic acid (*n* = 3). A two-tailed unpaired Student's *t* test was used to compare areas under the curve. Data are mean ± s.d. with *n* = 3 technical replicates per mouse. Bacterial colonies that were subjected to PAP testing are indicated in blue and orange (**c**, **d**). AMX amoxicillin, CLA clavulanic acid, VAN vancomycin, MIC minimum inhibitory concentration, c.f.u. colony-forming unit, AUC area under the curve. Source data are provided as a Source Data file.

Serial passaging of BPH0719 and DEN09 in 15 µg ml⁻¹ clavulanic acid with increasing concentrations of amoxicillin showed that their descendants also displayed increased resistance to other β-lactam antibiotics, including amoxicillin, clavulanic acid, penicillin G, and cefoxitin, but not to vancomycin and rifampicin (Fig. 5c, d). Importantly, growth rate measurements in antibiotic-free ISB revealed a decreasing growth rate in all the evolving populations, supporting that increased resistance to β-lactam antibiotics is associated with a high fitness cost (Fig. 5c, d).

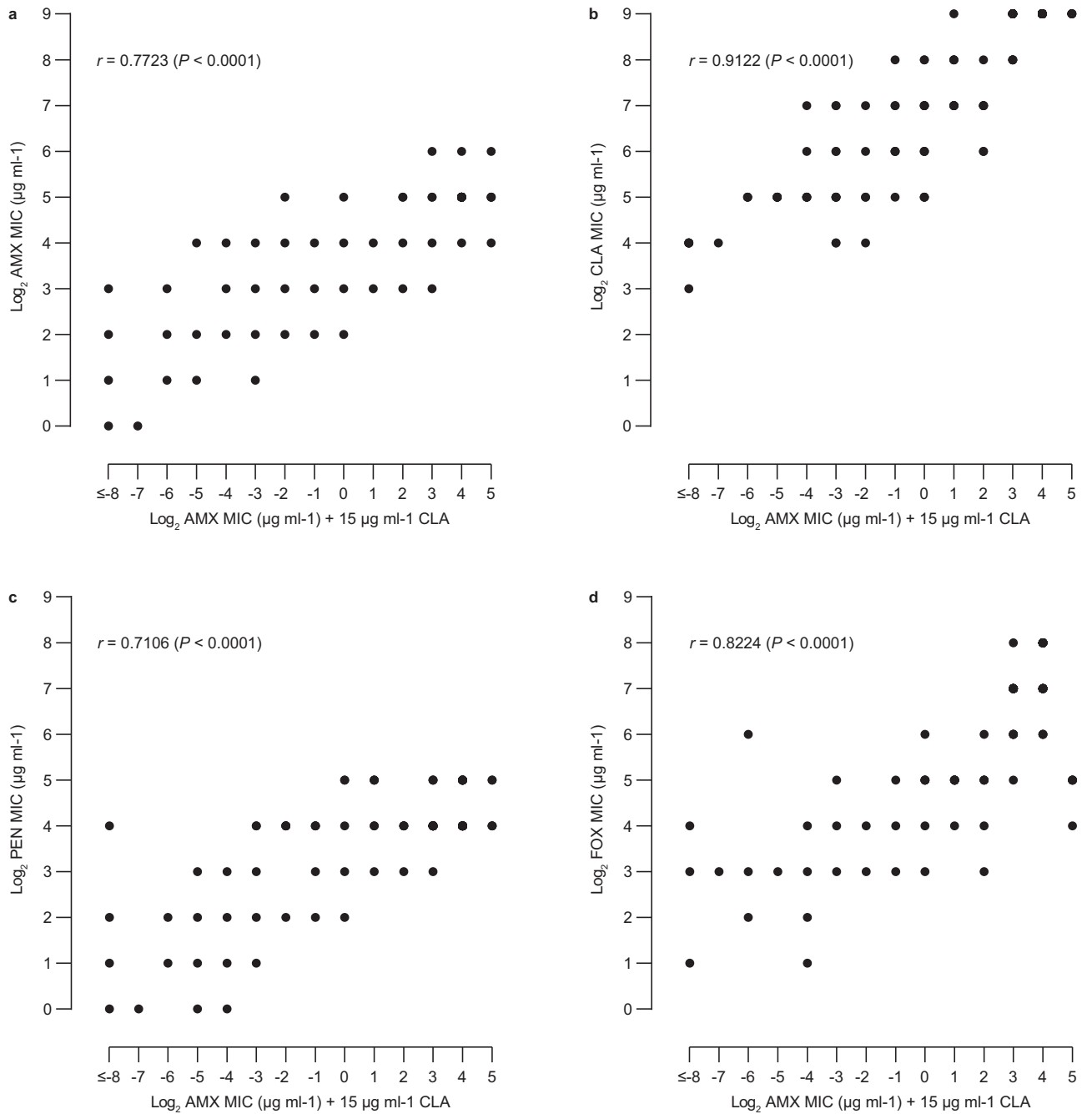

**Fig. 3 | Correlation between the minimum inhibitory concentrations of amoxicillin/clavulanic acid and other β-lactam antibiotics.** Comparison of the minimum inhibitory concentrations (MICs) of amoxicillin in the presence of 15 µg ml$^{-1}$ clavulanic acid and the MICs of amoxicillin alone (**a**), clavulanic acid alone (**b**), penicillin G alone (**c**), and cefoxitin alone (**d**). Two-tailed Spearman correlation tests were used to compare MICs of different antibiotics. AMX amoxicillin, CLA clavulanic acid, PEN penicillin G, FOX cefoxitin, *r* Spearman correlation coefficient. Source data are provided as a Source Data file.

To characterise adaptive trajectories leading to increased resistance to amoxicillin in the presence of 15 µg ml$^{-1}$ clavulanic acid, serially passaged descendants from six evolving populations of BPH0719 (*n* = 3) and DEN09 (*n* = 3) were isolated at various time points, coincident with increases in their MICs, and subjected to whole-genome sequencing. Genomic analysis identified 18 single-nucleotide polymorphisms (SNPs), of which 15 occurred in genes (eleven nonsynonymous, three synonymous, and one leading to a premature stop codon) and three in intergenic regions (Fig. 6a, b). Interestingly, we did not identify any mutations during the early stages of resistance development in the three BPH0719 populations, and none of the mutations were located in the *mecA*

promoter, PBP2a- and native PBP-encoding genes, or *blaZ*. Three genes encoding endoribonuclease RNase Y, c-di-AMP phospho-diesterase GdpP, and NAD(P)H-hydrate repair enzyme Nnr2 and one intergenic region (IR-1) were mutated in more than one of the evolving populations, indicating convergent evolution (Fig. 6a, b). Two of the mutated genes, *gdpP* encoding GdpP and *rpoB* encoding DNA-directed RNA polymerase subunit β, have previously been associated with mutation-driven development of resistance to β-lactam antibiotics in *S. aureus*[13–15]. To our knowledge, the other mutations have not been previously reported to be involved in resistance. We did not attempt to determine the effects of the individual mutations because our results suggest

**Table 1 | Amoxicillin susceptibility in the presence of 15 µg ml⁻¹ clavulanic acid among the 138 methicillin-resistant *Staphylococcus epidermidis* BPH0662 clone, ST2-mixed, ST5, and ST23 isolates from Australia, Denmark, and Germany**

| | Phenotype, %[a] | |
|---|---|---|
| | **Resistant** | **Susceptible** |
| Lineage | | |
| BPH0662 clone (*n* = 58) | 90% (*n* = 52) | 10% (*n* = 6) |
| ST2-mixed (*n* = 42) | 21% (*n* = 9) | 79% (*n* = 33) |
| ST5 (*n* = 6) | 17% (*n* = 1) | 83% (*n* = 5) |
| ST23 (*n* = 32) | 16% (*n* = 5) | 84% (*n* = 27) |
| Genotype | | |
| R2 (*n* = 77) | 78% (*n* = 60) | 22% (*n* = 17) |
| S2/S3 (*n* = 61) | 11% (*n* = 7) | 89% (*n* = 54) |

[a]Isolates were deemed phenotypically resistant or susceptible to amoxicillin in the presence of 15 µg ml⁻¹ clavulanic acid if they had minimum inhibitory concentrations above or at/below the empirical epidemiological cut-off (4.0 µg ml⁻¹), respectively.

that their clinical significance is doubtful. Firstly, we did not observe resistance development in the mouse infection model (Fig. 2d). Secondly, none of the mutations in RNase Y, GdpP, Nnr2, and IR-1 were present in the seven ST2-mixed, ST5, and ST23 isolates that were phenotypically resistant, but genotypically predicted to be susceptible (see above). On the other hand, we cannot rule out the possibility that resistance to penicillin/β-lactamase inhibitor combinations can also develop in vivo. Future studies should therefore seek to evaluate the in vivo risk and genetic basis of resistance development during prolonged therapy in animal models of persistent *S. epidermidis* infection.

**Anti-biofilm activity of amoxicillin/clavulanic acid**

Some *S. epidermidis* lineages such as ST2 are known to form strong biofilms on inert surfaces such as central venous catheters, prosthetic joints, and other indwelling medical devices[7]. Biofilms provide increased resilience toward host immune responses and antibiotics compared to their planktonic counterparts, and strong-biofilm-forming *S. epidermidis* lineages are therefore a more common cause of persistent infections[16]. In contrast, it has recently been shown that most *S. epidermidis* ST23 isolates lack the ability to form strong biofilms but instead have an increased ability to bind endothelial cells and persist in the bloodstream compared to ST2 isolates[17], which might explain why this species is also an important cause of non-biofilm-related infections. Assessment of biofilm formation by the 138 MRSE isolates, including the subset of 71 MRSE isolates that were phenotypically susceptible to amoxicillin/clavulanic, showed that BPH0662 clone and ST2-mixed isolates are much more likely to form moderate to strong biofilms than isolates belonging to ST23 as well as to ST5 (Fig. 4, Fig. 7a, b, and Supplementary Data 2). As expected, even high concentrations of amoxicillin/clavulanic acid (i.e., up to 128 µg ml⁻¹ amoxicillin in the presence of 15 µg ml⁻¹ clavulanic acid) failed to eradicate 6-h and 24-h biofilms formed by BPH0719, a phenotypically susceptible strong-biofilm-forming ST2-mixed isolate, although the amount of biofilm seemed to decline with increasing concentrations (Fig. 7c, d). It should be noted that the biofilm assay did not allow us to distinguish live from dead cells, and it is therefore possible that we have underestimated the anti-biofilm activity of amoxicillin/clavulanic acid.

**Potential synergies with other antibiotics**

β-lactam antibiotics alone display synergistic activity with vancomycin against vancomycin-susceptible MRSA strains as well as against vancomycin-intermediate *S. aureus* (VISA) and heterogeneous VISA strains[18], which underscores the potential usefulness of glycopeptides,

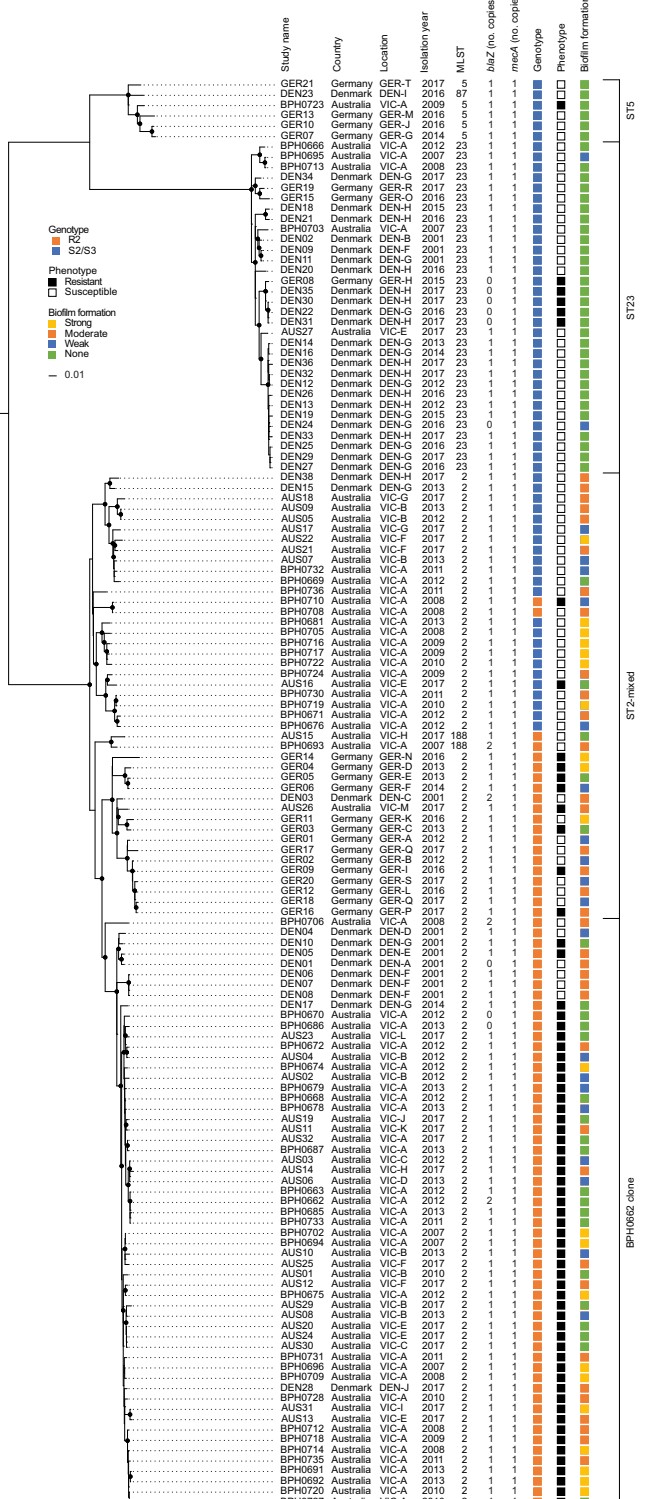

**Fig. 4 | Phylogenetic tree of 138 methicillin-resistant *Staphylococcus epidermidis* BPH0662 clone, ST2-mixed, ST5, and ST23 isolates from Australia, Denmark, and Germany.** The maximum-likelihood phylogeny was built from a core-genome single-nucleotide polymorphism (SNP) alignment (3102 SNPs) after putative recombination sites were removed. The tree was rooted at the midpoint. Branch support values above 90% are indicated by filled circles at the nodes. Isolates were deemed phenotypically resistant or susceptible to amoxicillin in the presence of 15 µg ml⁻¹ clavulanic acid if they had minimum inhibitory concentrations above or at/below the empirical epidemiological cut-off (4.0 µg ml⁻¹), respectively. The scale bar represents the number of nucleotide substitutions per variable site. MLST multilocus sequence type, AMX amoxicillin, CLA clavulanic acid.

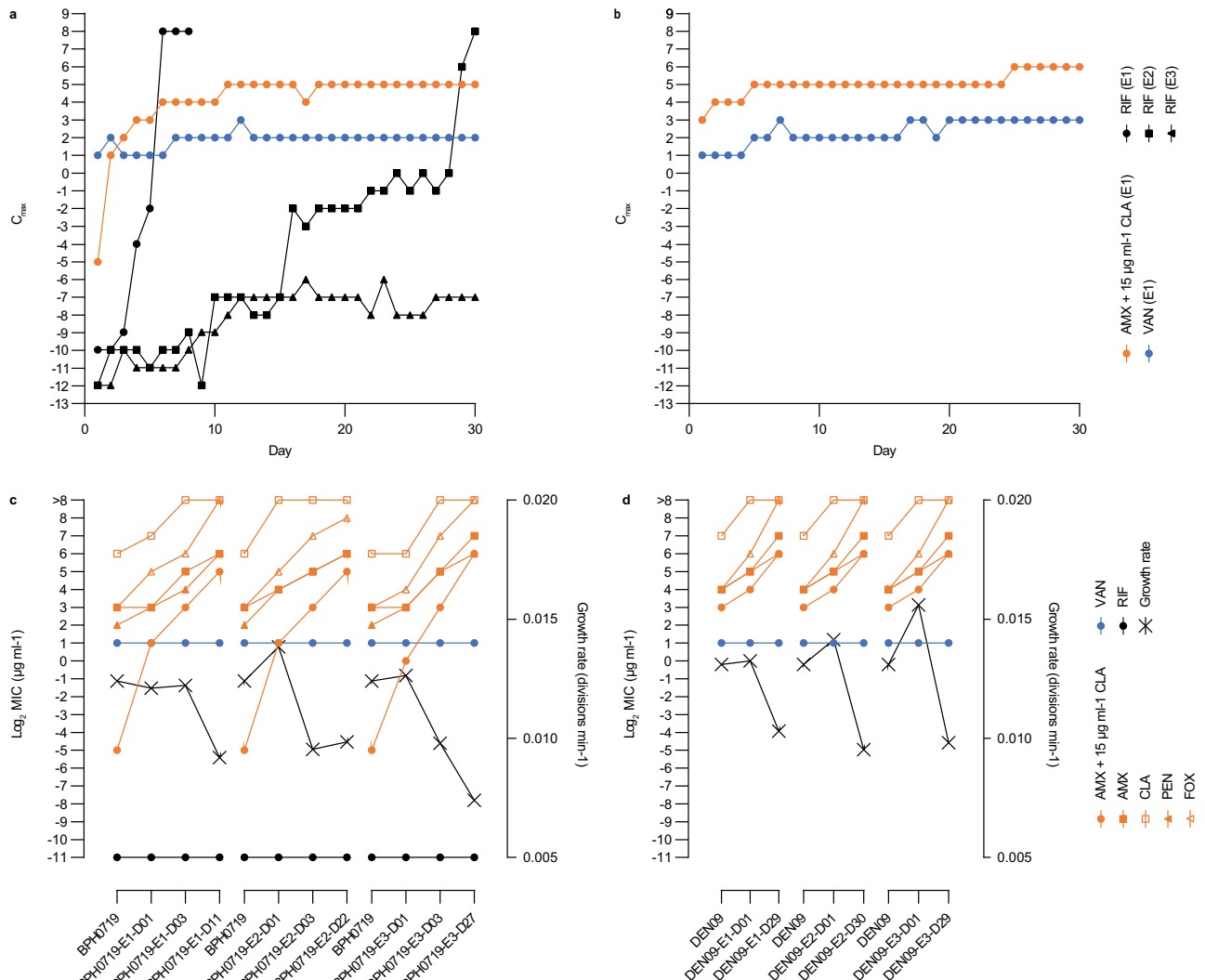

**Fig. 5 | Resistance development.** Serial passaging of BPH0719 (**a**) and DEN09 (**b**) in 15 μg ml⁻¹ clavulanic acid with increasing concentrations of amoxicillin and of vancomycin and rifampicin alone over 30 days. Representatives of up to six independent serial passaging experiments (E1–E6) are shown. Minimum inhibitory concentrations and growth rates of BPH0719 (**c**) and DEN09 (**d**) and their descendants during serial passaging in 15 μg ml⁻¹ clavulanic acid with increasing concentrations of amoxicillin and of vancomycin and rifampicin alone over 30 days. The descendants are named after the parent strain (e.g., BPH0719), the experiment (e.g., E1), and the day of isolation (e.g., D01). AMX amoxicillin, CLA clavulanic acid, PEN penicillin G, FOX cefoxitin, VAN vancomycin, RIF rifampicin, $C_{max}$ the highest drug concentrations that allowed growth after 24 h of incubation at 37 °C. Source data are provided as a Source Data file.

penicillins, and β-lactamase inhibitors as an alternative combination therapy for infections caused by multidrug-resistant staphylococci. As documented here, *S. epidermidis* is capable of developing increased resistance to different drug classes during long-term in vitro exposure, including β-lactam antibiotics, vancomycin, and rifampicin. In *S. aureus*, increased resistance to β-lactam antibiotics and vancomycin is driven by distinct mutational trajectories affecting different genes and regulatory pathways and is usually associated with a fitness cost in terms of reduced growth rates[15,19]. This suggests that combination treatment with these two drug classes will likely incur a fitness cost that is stronger than the fitness cost of either one of the individual drugs, thus decreasing the likelihood of resistance development.

Together, our results support that penicillin/β-lactamase inhibitor combinations could be a promising therapeutic candidate for short-term treatment of non-biofilm-related MRSE infections, while addition of other potentially potentiating drugs such as vancomycin might be used to counteract resistance development during prolonged therapy. Amoxicillin/clavulanic acid also had a partial effect on MRSE biofilms and thus might be useful in combination with rifampicin and other

drugs for treatment of medical device-related infections. However, additional studies are needed before penicillin/β-lactamase inhibitor combinations can be incorporated into clinical trials, including analyses to identify the optimal drug combinations and dosage regimens, assessments of their efficacy in in vitro and animal models of biofilm- and non-biofilm-related MRSE infections, and their effect on long-term population clearance and development of resistance and persistence.

## Methods
### Ethical statement
This research complies with all relevant ethical regulations. Animal experiments were approved by the Danish Animal Experiments Inspectorate (2016-15-0201-01049).

### Bacterial isolates and genomic analysis
We analysed the genomes of 227 *S. epidermidis* isolates originating from 96 institutions across 24 countries[4]. Sequence data comprised the closed genomes and plasmids of BPH0662 (GenBank accession no. NZ_LT571449.1, NZ_LT614820.1, NZ_LT571450.1, NZ_LT571451.1, and

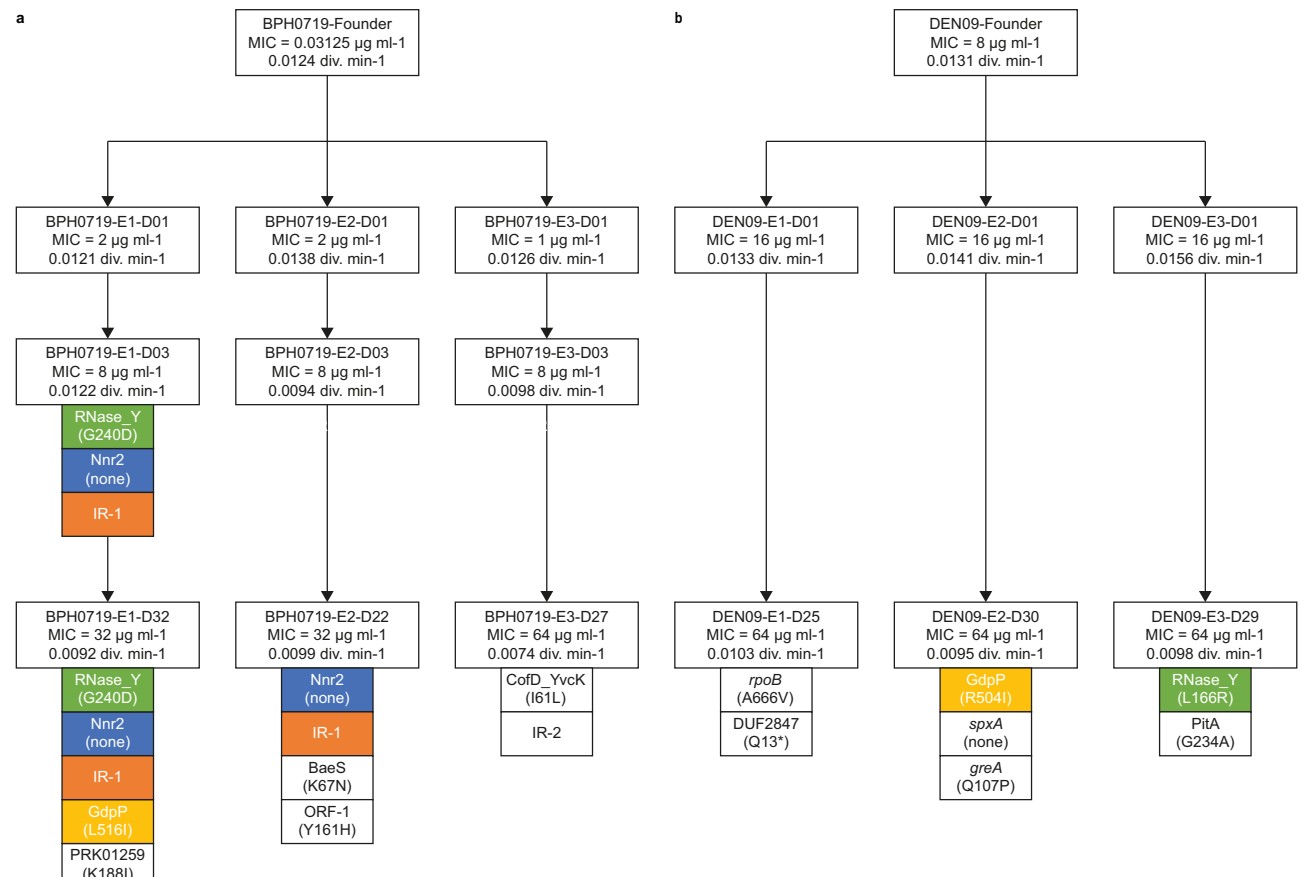

**Fig. 6 | Mutational trajectories.** Mutations in serially passaged descendants of BPH0719 (**a**) and DEN09 (**b**) associated with increased resistance to amoxicillin in the presence of 15 µg ml⁻¹ clavulanic acid. Descendants are named after the parent strain (e.g., BPH0719), the experiment (e.g., E1), and the day of isolation (e.g., D01). Amoxicillin/clavulanic acid minimum inhibitory concentrations and growth rates are indicated. Boxes show open reading frames (ORFs) and intergenic regions (IRs) containing single-nucleotide polymorphisms. ORFs and IRs that were altered in more than one of the evolving populations are indicated in coloured boxes. ORFs were named through RPS-BLAST searches against the NCBI Conserved Domain Database (version 3.20), setting the E-value cut-off at $1 \times 10^{-25}$. A single ORF did not produce hits and was therefore named ORF-1. Amino acid changes are included in parentheses. Stop codons are represented by asterisks. Three independent serial passaging experiments (E1–E3) are shown. Source data are provided as a Source Data file.

NZ_LR736240.1), PM221 (GenBank accession no. NZ_HG813242.1, NZ_HG813246.1, NZ_HG813245.1, NZ_HG813244.1, NZ_HG813243.1, and SEI (GenBank accession no. NZ_CP009046.1 and NZ_CP009047.1)) and Illumina paired-end reads of the remaining 224 *S. epidermidis* isolates (BioProjects PRJEB12090, PRJNA470534, and PRJNA470752). Draft genomes were de novo assembled with SPAdes (version 3.15)[20]. We used the *mecA* promoter and PBP2a-encoding gene in *S. aureus* strain COL (corresponding to nucleotide positions 39,643-41,718 in GenBank accession no. NC_002951), *blaZ* in *S. aureus* strain PC1 (corresponding to nucleotide positions 123-968 in GenBank accession no. M25252), and native PBP-encoding genes in *S. epidermidis* strain RP62A (*pbp1*, *pbp2*, *pbp3*, and SERP_RS06395 corresponding to nucleotide positions 741901-744228, 741901-744228, 1155781-1157871, and 1340351-1341256, respectively, in GenBank accession no. NC_002976) as queries in BLASTN searches against closed and draft *S. epidermidis* genomes, setting length match to 1.0 and similarity match to 0.9. The MUSCLE algorithm[21] was used to construct multiple sequence alignments. Insertion sequences elements were classified according to transposase gene similarity using BLASTN analysis with the ISfinder database[22]. Mapping of Illumina paired-end reads and SNP calling were carried out using NASP (version 1.0)[23] as follows: (1) Illumina paired-end reads were mapped against the closed genome of *S. epidermidis* isolate BPH0662 (GenBank accession no. NZ_LT571449.1) with the Burrows-Wheeler Alignment tool[24], (2) SNP calling was achieved using the GATK Unified

Genotyper[25,26], setting depth of coverage and unambiguously base calls to ≥10× and ≥90%, respectively, and ignoring insertions and deletions, and (3) SNPs contained in repeats were excluded using NUCmer[27,28]. Phages in BPH0662 were identified with PHASTER[29] and SNPs residing within these were manually removed from the alignment, whereafter Gubbins (version 2.3.4)[30] was used to remove recombination tracts. Phylogenetic reconstruction was carried out using the maximum-likelihood program PhyML (version 3.0) with a GTR model of nucleotide substitution[31,32]. The tree was rooted at the midpoint. Support for the nodes was assessed using aBayes[33].

**Antimicrobial susceptibility testing and empirical ECOFF determination**

MICs were determined in triplicate by the broth microdilution method in 96-well plates. Bacteria were individually grown overnight on blood agar plates (Oxoid) at 37 °C. Colonies were suspended in phosphate-buffered saline (PBS) to a 0.5 McFarland standard and diluted 1:100 in ISB (Oxoid). Microwells (Greiner-Bio-One) containing 50 µl of twofold dilutions of antibiotics (Sigma-Aldrich) in ISB (Oxoid) were inoculated with 50 µl of bacterial suspension. The final drug concentrations were 0.0039–128 µg ml⁻¹ for amoxicillin and penicillin G in the presence of 15 µg ml⁻¹ clavulanic acid, 0.0313–128 µg ml⁻¹ for amoxicillin and penicillin G, and 2–256 µg ml⁻¹ for clavulanic acid and cefoxitin. The lowest drug concentration that inhibited growth after 20 h of

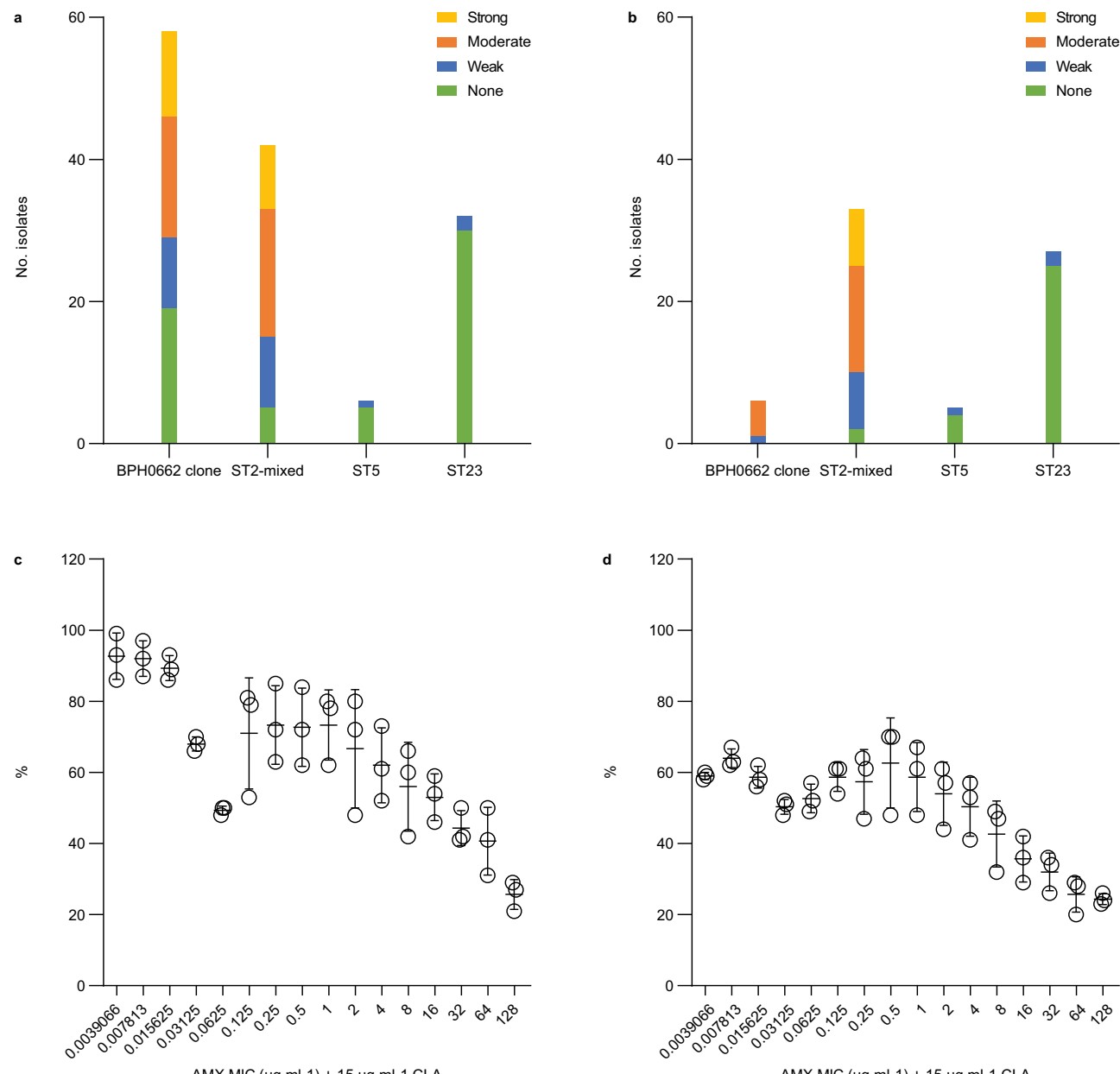

**Fig. 7 | Assessment of biofilm formation and anti-biofilm activity of amoxicillin in the presence of clavulanic acid.** Assessment of biofilm formation by the 138 methicillin-resistant *Staphylococcus epidermidis* (MRSE) isolates (**a**) and in a subset of 71 MRSE isolates that were phenotypically susceptible to amoxicillin/clavulanic (**b**). The ability to form biofilms (strong, moderate, weak, and none) was categorised using a previously described scoring system[34]. Dose-response effect of amoxicillin in the presence of 15 µg ml⁻¹ clavulanic acid against BPH0719 allowed to form biofilms for 6-h (**c**) and 24-h (**d**). The *y* axes show the amounts of biomass as percentages of the results for the matching control (**c**, **d**). Data are mean ± s.d. with *n* = 3 technical replicates. AMX amoxicillin, CLA clavulanic acid. Source data are provided as a Source Data file.

incubation at 35 °C was interpreted as the MIC. The empirical ECOFF for amoxicillin in the presence of 15 µg ml⁻¹ clavulanic acid was determined using a custom R code (https://doi.org/10.5281/zenodo. 8344500)[34].

### Resistance development

The initial MICs of amoxicillin (Sigma-Aldrich) in the presence of 15 µg ml⁻¹ clavulanic acid (Sigma-Aldrich), vancomycin (Sigma-Aldrich), and rifampicin (Sigma-Aldrich) were determined in triplicate by inoculating microwells (Greiner-Bio-One) containing 100 µl of twofold dilutions of antibiotics in ISB (Oxoid) with 100 µl of bacterial suspension prepared in the same way as for broth microdilution. Cultures from the highest drug concentrations that allowed growth after 24 h of

incubation at 37 °C were adjusted back to a 0.5 McFarland standard and used to inoculate fresh microwells containing antibiotics. This procedure was repeated daily for 30 days. The final drug concentrations were 0.0156–256 µg ml⁻¹ for amoxicillin in the presence of 15 µg ml⁻¹ clavulanic acid, 0.5–128 µg ml⁻¹ for vancomycin, and 0.000125–256 µg ml⁻¹ for rifampicin. Cultures that grew at drug concentrations greater than the MIC were stored in 25% (v/v) glycerol at −80 °C for further analysis. Serial passaging experiments were performed with three to six biological replicates.

To characterise adaptive trajectories leading to increased resistance to amoxicillin in the presence of 15 µg ml⁻¹ clavulanic acid, serially passaged descendants displaying increased MICs were subjected to whole-genome sequencing on a NextSeq platform (Illumina) with

2 × 151 bp. Illumina paired-end reads of BPH0719 and DEN09 were retrieved from Lee et al. (BioProjects PRJEB12090 and PRJNA470752)[4]. Draft genomes were de novo assembled with SPAdes (version 3.15)[20] and annotated with the NCBI Prokaryotic Genome Annotation Pipeline. Mapping of Illumina paired-end reads and SNP calling were carried out as described above, except that Illumina paired-end reads were mapped against the annotated draft genomes of BPH0719 or DEN09. The MUSCLE algorithm[21] was used to construct multiple sequence alignments of open reading frames (ORFs) and intergenic regions containing SNPs. ORFs were further annotated using RPS-BLAST searches against the NCBI Conserved Domain Database (version 3.20), setting the E-value cut-off at $1 \times 10^{-25}$. Short-read sequence data and annotated draft genomes generated in this study are available in the European Nucleotide Archive/NCBI Sequence Read Archive under BioProject PRJNA898869 and the accession numbers are provided in Supplementary Table 2. MICs were determined by the broth microdilution method in 96-well plates as described above. For growth rate measurements, bacteria were individually grown overnight on blood agar plates (Oxoid) at 37 °C. Colonies were suspended in PBS to a 0.5 McFarland standard and diluted 1:100 in ISB (Oxoid). The suspensions were used to start two to three 300 μl cultures that were grown for 24 h at 37 °C in a Bioscreen C incubator (Oy Growth Curves) with continuous shaking and automatic $OD_{600}$ plate readings every 30 min. Doubling times were calculated using GraphPad Prism (version 8.3) (GraphPad Software). The growth rate (i.e., the number of divisions per unit of time) was expressed as the reciprocal of the doubling time. Experiments were performed with three biological replicates.

## MPC determination

The initial MICs of amoxicillin (Sigma-Aldrich) in the presence of 15 μg ml⁻¹ clavulanic acid (Sigma-Aldrich) were determined in triplicate by the agar dilution method. ISA plates (Oxoid) containing 15 μg ml⁻¹ clavulanic acid and twofold dilutions of amoxicillin (range, 0.0313–16 μg ml⁻¹) were spotted with 2 μl of bacterial suspension prepared in the same way as for broth microdilution. The lowest drug concentration that inhibited growth after 18 h of incubation at 35 °C was interpreted as the MIC. MPCs were determined with three biological replicates by plating $10^9$ cells onto ten ISA plates (Oxoid) containing 15 μg ml⁻¹ clavulanic acid and twofold dilutions of amoxicillin (range, 0.125–32 μg ml⁻¹). After 48 h of incubation at 37 °C, the plates were examined for colonies.

## Mouse infection model

Animal experiments were performed in the animal facility at Statens Serum Institut using 6–8 weeks old female NMRI mice (Envigo). Mice were housed individually in ventilated cages at a constant temperature (22 ± 2 °C) and relative humidity (55 ± 10%), with a 12:12 h light:dark cycle (light on 06.00–18.00 h). Food and water were available ad libitum. Mice were rendered neutropenic by two intraperitoneal injections (0.5 ml) of cyclophosphamide (Sendoxan; Baxter), administered four days (200 mg kg−1) and one day (100 mg kg⁻¹) before the experiment, respectively. Approximately 1 h before the experiment, mice were treated orally (45 μl) with 30 mg kg⁻¹ ibuprofen (Nurofen Junior; Reckitt Benckiser Healthcare) for pain relief. Mice were inoculated intramuscularly in the left thigh (50 μl) with around $8.5 \times 10^7$ colony-forming units. After 1 h, mice ($n = 6$ per group) were treated subcutaneously with a single dose of vehicle alone, 100 μg kg⁻¹ or 250 μg kg⁻¹ amoxicillin alone (Multimox; Bela-Pharm) or in combination with 20 mg kg⁻¹ or 50 mg kg⁻¹ clavulanic acid (Augmentin; Beecham Group), or 40 mg kg⁻¹ vancomycin (Fresenius Kabi). Mice were sacrificed 1 h postinfection (i.e., start of treatment) or 5 h postinfection (i.e., 4 h after treatment), their left thigh aseptically excised, suspended in 5 ml of sterile saline, homogenised for 3 min using a Xiril Dispomix Drive homogenizer (Medic Tools), serially tenfold diluted in sterile saline containing 0.1% Triton X-100 (Sigma-Aldrich), plated in

duplicate on blood agar (SSI Diagnostica), and incubated overnight at 37 °C for enumeration. Mice were monitored for signs of pain or distress throughout the experiment. Experiments were performed with six mice per group.

## PAP-AUC

Bacterial colonies from mice treated with the vehicle alone ($n = 3$) or 250 mg kg⁻¹ amoxicillin in combination with 50 mg kg⁻¹ clavulanic acid ($n = 3$) were subjected to PAP testing. Bacteria were individually grown overnight in Tryptone Soy Broth (TSB) at 37 °C, serially tenfold diluted in sterile saline to $10^{-6}$, plated on ISA plates (Oxoid) containing 15 μg ml⁻¹ clavulanic acid alone or ISA plates (Oxoid) containing 15 μg ml⁻¹ clavulanic acid and twofold dilutions of amoxicillin (range, 1–128 μg ml⁻¹), and incubated overnight at 37 °C for enumeration. AUCs were calculated using GraphPad Prism (version 8.3) (GraphPad Software). Experiments were performed with three technical replicates per mouse.

## Biofilm assays

Biofilm formation was evaluated in 96-well plates by measuring the absorbance of Gram's safranin. In brief, bacterial cells harvested from overnight culture were suspended in TSB to a turbidity equivalent to a 0.5 McFarland standard. Nunclon Delta Surface microwells (Thermo-Fisher Scientific) containing 200 μl of bacterial suspension were incubated for 24 h at 37 °C, washed three times with PBS, and then stained by 10 min of incubation at room temperature with 200 μl of a Gram's safranin solution (Sigma-Aldrich). Biofilms were washed with distilled water to remove excess dye and allowed to dry for 1 h at room temperature. The dye fixed to the biofilm was resolubilized by the addition of 200 μl of 33% acetic acid and incubation at room temperature for 1 h without shaking. Gram's safranin absorbance was measured at a wavelength of 530 nm in a FLUOstar Omega microplate reader (BMG LABTECH). *S. epidermidis* strains ATCC 12228 and 1457 were used as negative and positive controls, respectively. Experiments were performed with three technical replicates. The ability to form biofilms (strong, moderate, weak, and none) was categorised using a previously described scoring system[35].

Antibiotic activity was evaluated against 6-h and 24-h biofilms that were prepared as described above. When the desired maturity was reached, the biofilm culture medium was removed and immediately replaced by 200 μl of ISB alone (control) or 200 μl of ISB containing 15 μg ml⁻¹ clavulanic acid and amoxicillin at increasing concentrations. The final drug concentrations were 0.0039–128 μg ml⁻¹. Biofilms were reincubated for 24 h at 37 °C, and then Gram's safranin absorbance was measured as described above. To correct for growth of the biofilm during incubation, all data are expressed as percentages of the results for the matching control. Experiments were performed with three technical replicates.

## Statistics

Statistical analyses were performed using GraphPad Prism (version 8.3) (GraphPad Software). We used a two-tailed unpaired Student's $t$ test to compare AUCs and two-tailed Spearman correlation tests to compare MICs of different antibiotics.

## Reporting summary

Further information on research design is available in the Nature Portfolio Reporting Summary linked to this article.

## Data availability

*S. epidermidis* short-read sequence data and annotated genomes generated in this study have been deposited in the European Nucleotide Archive/NCBI Sequence Read Archive under BioProject PRJNA898869 and the accession numbers are provided in Supplementary Table 2. Sequence data from other sources comprised the

closed genomes and plasmids of BPH0662 (GenBank accession no. NZ_LT571449.1, NZ_LT614820.1, NZ_LT571450.1, NZ_LT571451.1, and NZ_LR736240.1), PM221 (GenBank accession no. NZ_HG813242.1, NZ_HG813246.1, NZ_HG813245.1, NZ_HG813244.1, NZ_HG813243.1, and SEI (GenBank accession no. NZ_CP009046.1 and NZ_CP009047.1), Illumina paired-end reads of the remaining 224 *S. epidermidis* isolates (BioProjects PRJEB12090, PRJNA470534, and PRJNA470752), the *mecA* promoter and PBP2a-encoding gene in *S. aureus* strain COL (corresponding to nucleotide positions 39,643-41,718 in GenBank accession no. NC_002951), *blaZ* in *S. aureus* strain PC1 (corresponding to nucleotide positions 123-968 in GenBank accession no. M25252), and native PBP-encoding genes in *S. epidermidis* strain RP62A (*pbp1*, *pbp2*, *pbp3*, and SERP_RS06395 corresponding to nucleotide positions 741901-744228, 741901-744228, 1155781-1157871, and 1340351-1341256, respectively, in GenBank accession no. NC_002976). Information about the 227 *S. epidermidis* isolates used in this study is provided in Supplementary Data 1. Additional information about the 138 MRSE BPH0662 clone, ST2-mixed, ST5, and ST23 isolates from Australia, Denmark, and Germany is provided in Supplementary Data 2. The PhyML tree file for the 138 MRSE BPH0662 clone, ST2-mixed, ST5, and ST23 isolates from Australia, Denmark, and Germany is provided in Newick format in Supplementary Data 3. Source data for Figs. 1–3 and Figs. 5–7 are provided with this paper.

## Code availability
The custom R code for empirical ECOFF determination is available through Zenodo (https://doi.org/10.5281/zenodo.8344500)[34].

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

## Acknowledgements

This work was supported by the Wellcome Trust under Grant 220540/Z/20/A (E.M.H.). We thank Frederikke Rosenborg Petersen and Emilie Due Jensen for technical assistance during the animal experiments.

## Author contributions

X.B., C.L.R., E.M.H., A.R.L., M.A.H., and J.L. initiated and designed the study. J.Y.H.L., B.P.H., M.D.B., and B.S. provided *S. epidermidis* isolates. X.B., C.L.R., and L.M.C. carried out laboratory experiments. O.R. determined the empirical ECOFF. J.L. performed whole-genome sequencing and bioinformatics analyses. C.V.L designed and conducted animal experiments. J.L. wrote the manuscript with considerable inputs from X.B., C.L.R., E.M.H., A.R.L., and M.A.H. All authors reviewed the manuscript.

## Competing interests

The authors declare no competing interests.
