## [Peer Review File · Nature Communications]

REVIEWER COMMENTS

Reviewer #1 (Remarks to the Author):

The manuscript by Ba et al reports the existence of susceptibility to betalactam/betalactamase inhibitor combinations in the setting of MRSE. This follows from previous reports of this phenomenon in MRSA strains. The work here documents the existence of this phenotype in clinical isolates, shows that susceptibility is also associated with clinical success in a mouse thigh model, shows that this phenotype can be established after serial passage, and shows that susceptibility is associated with some specific genomic changes.

Major critiques:

1. The mouse model is not really reflective of the most common kinds of infections caused by Staph epi, and it therefore does not establish a strong basis for the use of betalactam/betalactamase inhibitor combinations in clinical practice. While Staph epi occasionally causes skin/soft tissue infections in compromised individuals, it is much more of a problem for indwelling catheters and hardware.
2. The genetic changes associated with serial passages are not tested using genetic approaches for their role in susceptibility, and they therefore remain associations. This also limits the ability of the authors to assess the mechanisms of susceptibility.

Minor concerns:

1. Lines 62 and 82: It is not generally accepted that SCCmec elements in Staph aureus originated in Staph epi. See Miragaia 2018 for a review.
2. Line 99: Why is it important to point out that amox/clav is “already available for intravenous injection”? In fact ampicillin sulbactam is the IV drug of choice analogous to amox/clav.
3. Among site rate variation model is not reported for phylogenetic analysis.

Reviewer #2 (Remarks to the Author):

Thank you for the opportunity to review “Cryptic susceptibility to penicillins and β -lactamase inhibitors in emerging multidrug-resistant, hospital-adapted Staphylococcus epidermidis lineages” by Ba and colleagues. The authors present an extensive analysis of *S. epidermidis* susceptibility to penicillins and β -lactamase inhibitors. The foundation for the study was a previously published population genomic

analysis of 227 *S. epidermidis* isolates from 24 countries. The authors re-analyse the genomic data specifically accessing genomic determinants of antibiotic resistance. They obtained 138 of the isolate and performed a comprehensive phenotypic analysis of antibiotic susceptibilities. They then advance these studies with in vivo experiments as well as experimental evolution studies. Their results elucidate the genomics of resistance in *S. epidermidis* and inform future trials of antibiotic treatment. Overall, I thoroughly enjoyed this study. It was well-conceived and executed. It is clearly written, and a tremendous amount of supplemental material is made available for detailed review of the findings. In particular, the experimental evolution component was a great way to tie in the observational results and resolve mechanisms of resistance. It would be great to see this component expanded in future studies to include more strains. Taken together, I feel the study is of broad interest to the field and only have minor comments below.

1) I would suggest for the paragraph starting on line 67 that the authors more clearly delineate the findings from Lee et al and the present work. I believe everything from line 69 on was new analysis, but the authors should clarify.

2) It's a minor stylistic suggestion, but I prefer the paragraphs to not start with a citation (e.g., Harrison et al., Lee et al.). In several places I thought the syntax would flow better without explicitly stating the study. I thought it detracted from the new analysis that was conducted in the present study.

3) Regarding the main figures, it may just be how they are appearing imbedded in the text, but some of the legends are hard to read.

4) Supplementary figure 1 is sufficient as is, but I thought that it could be considered as a main figure with some minor addition of a heatmap, tip shapes, and colour. In particular, it is a great way to visualize the phylogenetic relationship between the genotype and AMX MIC values.

Reviewer #3 (Remarks to the Author):

Key Results:

This study was based on a previous study that described collateral sensitivity to penicillin in the presence of clavulanate in MRSA (Harrison et al, ref 6), and the authors here showed that MDR *S. epidermidis* (MRSE) also display such property and its susceptibility to amoxicillin/clavulanate demonstrated in a mouse thigh model of infection. A collection of *S. epidermidis* strains with well-defined genotypes was included (Lee et al, ref 4) but the effect was heterogenous with varying susceptibilities, while increased resistance was also observed with serial passaging of these MRSE

strains. A number of new mutations/genes were identified and associated with the resistance development but these SNPs individually were not further elucidated.

Validity:

The results were carefully documented to illustrate the reduction of MICs with b-lactam/clavulanate, strain lineage/types, and resistance mechanisms and mutations of *mecA* promoter, a.a. substitutions in PBP2a or of other genes. These latter were identical to that described in MRSA (ref 6). Further laboratory studies using site-directed mutagenesis may be conducted to validate or elucidate the relevance of the new mutations identified in the subsequent increasing resistant strains.

Significance:

The study highlights a possible option of using penicillin-clavulanate for the treatment of MRSE infections, as proposed in the previous publication (Harrison et al) but with MRSA. However, the effect with MRSE is quite heterogeneous (ie 'cryptic susceptibility') depending on the individual strains under variable background resistance mechanisms eg reduced *mecA* expression with altered aa in *pbp2a* and not lineage related, and compromised by the development of resistance with serial passaging in presence of clavulanate. This is probably not surprising given that *S. epidermidis* characteristically is associated with catheter/implant and biofilm infections that will select for resistance in the presence of these antibiotics, as well as slower growth with variants development. The clinical translational value is likely to be quite limited, given that the increased amx/cla susceptibility has to be based on many background prior premises of conditions (and possibly other unknown SNPs/genes).

Data and methodology

The methods are detailed and there is also detailed documentation of the results.

However, the way the description and results are presented makes the manuscript quite difficult to digest.

Suggested improvements and clarity and context

There are a lot of results and figures generated based on the previous collection of strains (from reference 4). The manuscript might be easier to read by just describing the strains that became susceptible to 'amx/clavulanate' to demonstrate and highlight the key findings. The manuscript could be shortened and more succinct. The remainder/ complete set of results could just go to supplementary files. While the whole collection of strains is delineated into genotypes/clades, it may not be very relevant except that the effect is observed from strains isolated from laboratories in several countries.

Reviewer #1 (Remarks to the Author):

The manuscript by Ba et al reports the existence of susceptibility to betalactam/betalactamase inhibitor combinations in the setting of MRSE. This follows from previous reports of this phenomenon in MRSA strains. The work here documents the existence of this phenotype in clinical isolates, shows that susceptibility is also associated with clinical success in a mouse thigh model, shows that this phenotype can be established after serial passage, and shows that susceptibility is associated with some specific genomic changes.

Major critiques:

1. The mouse model is not really reflective of the most common kinds of infections caused by Staph epi, and it therefore does not establish a strong basis for the use of betalactam/betalactamase inhibitor combinations in clinical practice. While Staph epi occasionally causes skin/soft tissue infections in compromised individuals, it is much more of a problem for indwelling catheters and hardware.

RESPONSE 1.1: Thank you for raising this point. Some biofilm-forming *S. epidermidis* lineages such as ST2 are indeed a common cause of medical device-related infections (see Ref. 7 and Ref. 26), whereas other lineages such as ST23 have acquired alternative mechanisms to persist in the bloodstream (see Ref. 17). Most of the phenotypically susceptible *S. epidermidis* isolates belonged to ST23, and we therefore decided to use animal models of non-biofilm-related *S. epidermidis* infections. However, we failed to establish persistent infection in a mouse bacteraemia model, which is consistent with the fact that planktonic *S. epidermidis* cells are rapidly cleared from the bloodstream of both naïve and immunocompromised mice (doi:10.1128/IAI.01472-15). In addition, it has been shown that immunosuppression alone is not sufficient to establish *S. epidermidis* bloodstream infection in a rat jugular catheter model (doi:10.1128/IAI.02177-14). These limitations are the reasons why we decided to try the same mouse thigh model as Harrison et al. (see Ref. 6) used with success to demonstrate in vivo efficacy of amoxicillin/clavulanic acid against an MRSA USA300 strain. We have specifically modified the last sentence to emphasize the importance of using clinically more relevant animal models in the next drug discovery phase (see l. 244-249).

Prompted by the comment on medical device-related infections, we screened the 138 MRSE isolates for biofilm formation, which confirmed that ST2 isolates belonging to the BPH0662 clone and ST2-mixed are much more likely to produce biofilms than isolates belonging to ST23 and ST5. We also assessed the anti-biofilm activity of amoxicillin/clavulanic acid against BPH0719, a phenotypically susceptible strong-biofilm-forming ST2-mixed isolate. The results showed that even high concentrations of amoxicillin/clavulanic acid failed to eradicate 6-h and 24-h biofilms formed by BPH0719, although it should be noted that the amount of biofilm seemed to decline with increasing concentrations. These findings are consistent with the fact that biofilms provide increased resilience toward host immune responses and antibiotics. We have included these experiments and results in the revised manuscript (see l. 207-225, l. 238-249, l. 357-377, and Fig. 7).

2. The genetic changes associated with serial passages are not tested using genetic approaches for their role in susceptibility, and they therefore remain associations. This also limits the ability of the authors to assess the mechanisms of susceptibility.

RESPONSE 1.2: We did not attempt to construct isogenic mutants to test the role of individual genetic changes in resistance development because our results suggest that their clinical significance is doubtful (see l. 198-202). Moreover, the results will not change the clinically more important conclusion that both isolates are capable of developing phenotypic resistance under in vitro conditions. We have specifically modified the last parts of the relevant paragraph to explain our standpoint and emphasize that other studies in animal models of persistent *S. epidermidis*

infection are needed evaluate the in vivo risk and genetic basis of resistance development during prolonged therapy (see l. 198-206).

Minor concerns:

1. Lines 62 and 82: It is not generally accepted that SCCmec elements in Staph aureus originated in Staph epi. See Miragaia 2018 for a review.
RESPONSE 1.3: We do not challenge the view that the first primordial SCCmec elements probably evolved and were assembled in other CoNS than S. epidermidis. However, it is equally accepted that some of the predominant SCCmec elements in S. aureus were acquired from S. epidermidis in the “antibiotic era” (while still recognising that some parts might have originated from other CoNS before their transfer to S. epidermidis) (see Ref. 8).
2. Line 99: Why is it important to point out that amox/clav is “already available for intravenous injection”? In fact ampicillin sulbactam is the IV drug of choice analogous to amox/clav.
RESPONSE 1.4: We acknowledge the existence of other penicillin/ β -lactamase inhibitor combinations than amoxicillin/clavulanic acid for intravenous use and have therefore deleted the sentence.
3. Among site rate variation model is not reported for phylogenetic analysis.
RESPONSE 1.5: This is because GTR, which assumes that rates are homogeneous across sites, was predicted to be the best model by Smart Model Selection in PhyML:

Model	Decoration	K	Lik	AIC	BIC
GTR		281	-23441,3	47444,67	49141,85
GTR	+I	282	-23441,4	47446,73	49149,95
GTR	+G	282	-23442,4	47448,87	49152,1
TN93		278	-23643,9	47843,8	49522,87
GTR	+G+I	283	-23645,3	47856,58	49565,85

Reviewer #2 (Remarks to the Author):

Thank you for the opportunity to review “Cryptic susceptibility to penicillins and β -lactamase inhibitors in emerging multidrug-resistant, hospital-adapted *Staphylococcus epidermidis* lineages” by Ba and colleagues. The authors present an extensive analysis of *S. epidermidis* susceptibility to penicillins and β -lactamase inhibitors. The foundation for the study was a previously published population genomic analysis of 227 *S. epidermidis* isolates from 24 countries. The authors re-analyse the genomic data specifically accessing genomic determinants of antibiotic resistance. They obtained 138 of the isolate and performed a comprehensive phenotypic analysis of antibiotic susceptibilities. They then advance these studies with in vivo experiments as well as experimental evolution studies. Their results elucidate the genomics of resistance in *S. epidermidis* and inform future trials of antibiotic treatment. Overall, I thoroughly enjoyed this study. It was well-conceived and executed. It is clearly written, and a tremendous amount of supplemental material is made available for detailed review of the findings. In particular, the experimental evolution component was a great way to tie in the observational results and resolve mechanisms of resistance. It would be great to see this component expanded in future studies to include more strains. Taken together, I feel the study is of broad interest to the field and only have minor comments below.

- 1) I would suggest for the paragraph starting on line 67 that the authors more clearly delineate the findings from Lee et al and the present work. I believe everything from line 69 on was new analysis, but the authors should clarify.
RESPONSE 2.1: Thank you for this very useful suggestion. As a consequence, we have divided the main text into a section headed “Introduction” and a section headed “Results and discussion”.
- 2) It’s a minor stylistic suggestion, but I prefer the paragraphs to not start with a citation (e.g., Harrison et al., Lee et al.). In several places I thought the syntax would flow better without explicitly stating the study. I thought it detracted from the new analysis that was conducted in the present study.
RESPONSE 2.2: We have changed the syntax throughout the revised manuscript.
- 3) Regarding the main figures, it may just be how they are appearing imbedded in the text, but some of the legends are hard to read.
RESPONSE 2.3: We are sorry for the inconvenience and have uploaded high-resolution versions of each figure in TIFF format.
- 4) Supplementary figure 1 is sufficient as is, but I thought that it could be considered as a main figure with some minor addition of a heatmap, tip shapes, and colour. In particular, it is a great way to visualize the phylogenetic relationship between the genotype and AMX MIC values.
RESPONSE 2.4: Thank you for this suggestion. We have made a new version of this figure and included it in the main manuscript (see **Fig. 4**).

Reviewer #3 (Remarks to the Author):

Key Results:

This study was based on a previous study that described collateral sensitivity to penicillin in the presence of clavulanate in MRSA (Harrison et al, ref 6), and the authors here showed that MDR *S. epidermidis* (MRSE) also display such property and its susceptibility to amoxicillin/clavulanate demonstrated in a mouse thigh model of infection. A collection of *S. epidermidis* strains with well-defined genotypes was included (Lee et al, ref 4) but the effect was heterogenous with varying susceptibilities, while increased resistance was also observed with serial passaging of these MRSE strains. A number of new mutations/genes were identified and associated with the resistance development but these SNPs individually were not further elucidated.

Validity:

The results were carefully documented to illustrate the reduction of MICs with b-lactam/clavulanate, strain lineage/types, and resistance mechanisms and mutations of *mecA* promoter, a.a. substitutions in PBP2a or of other genes. These latter were identical to that described in MRSA (ref 6). Further laboratory studies using site-directed mutagenesis may be conducted to validate or elucidate the relevance of the new mutations identified in the subsequent increasing resistant strains.

RESPONSE 3.1: We did not attempt to construct isogenic mutants to test the role of individual genetic changes in resistance development because our results suggest that their clinical significance is doubtful (see l. 198-202). Moreover, the results will not change the clinically more important conclusion that both isolates are capable of developing phenotypic resistance under in vitro conditions. We have specifically modified the last parts of the relevant paragraph to explain our standpoint and emphasise that other studies in animal models of persistent *S. epidermidis* infection are needed evaluate the in vivo risk and genetic basis of resistance development during prolonged therapy (see l. 198-206).

Significance:

The study highlights a possible option of using penicillin-clavulanate for the treatment of MRSE infections, as proposed in the previous publication (Harrison et al) but with MRSA. However, the effect with MRSE is quite heterogenous (ie 'cryptic susceptibility') depending on the individual strains under variable background resistance mechanisms eg reduced *mecA* expression with altered aa in *pbp2a* and not lineage related, and compromised by the development of resistance with serial passaging in presence of clavulanate. This is probably not surprising given that *S. epidermidis* characteristically is associated with catheter/implant and biofilm infections that will select for resistance in the presence of these antibiotics, as well as slower growth with variants development. The clinical translational value is likely to be quite limited, given that the increased amx/clo susceptibility has to be based on many background prior premises of conditions (and possibly other unknown SNPs/genes).

RESPONSE 3.2: We would argue that there is a rather strong association between the phylogenetic position of the MRSE isolates and their susceptibility/resistance to amoxicillin-clavulanic acid in that most MRSE ST5 and ST23 and early-branching ST2-mixed isolates are phenotypically susceptible, whereas most of the late-branching MRSE ST2-mixed and BPH0662 clone isolates are resistant. We agree that there are several microevolutionary deviations from this general phenotypic pattern but this is a common phenomenon that also applies to *S. epidermidis* resistance against the last-resort antibiotics vancomycin and rifampicin (see Ref. 4).

We appreciate the concerns regarding the development of resistance against amoxicillin/clavulanic acid under in vitro conditions. However, our results suggest that their clinical significance is doubtful (see l. 198-202). Moreover, *S. epidermidis* is also capable of developing resistance against the last-resort antibiotics vancomycin and rifampicin, which suggests that this is a general problem.

Some biofilm-forming *S. epidermidis* lineages such as ST2 are indeed a common cause of medical device-related infections, whereas other lineages such as ST23 have acquired alternative mechanisms to persist in the bloodstream. We have now screened the 138 MRSE isolates for biofilm formation, which

confirmed that ST2 isolates belonging to the BPH0662 clone and ST2-mixed are much more likely to produce biofilms than isolates belonging to ST23 and ST5. We also assessed the anti-biofilm activity of amoxicillin/clavulanic acid against BPH0719, a phenotypically susceptible strong-biofilm-forming ST2-mixed isolate. The results showed that even high concentrations of amoxicillin/clavulanic acid failed to eradicate 6-h and 24-h biofilms formed by BPH0719, although it should be noted that the amount of biofilm seemed to decline with increasing concentrations. These findings are consistent with the fact that biofilms provide increased resilience toward host immune responses and antibiotics. We have included these experiments and results (see l. 207-225, l. 238-249, l. 357-377, and Fig. 7).

In conclusion, we still believe that penicillin/ β -lactamase inhibitor combinations could be a promising therapeutic candidate for short-term treatment of non-biofilm-related multidrug-resistant MRSE infections, while addition of other potentially potentiating drugs such as vancomycin might be used to counteract resistance development during prolonged therapy. Amoxicillin/clavulanic acid also had a partial effect on *S. epidermidis* biofilms and thus might be useful in combination with rifampicin and other drugs for treatment of medical device-related infections. However, we do acknowledge that additional studies are needed before penicillin/ β -lactamase inhibitor combinations can be incorporated into clinical trials, including analyses to identify the optimal drug combinations and dosage regimens, assessments of their efficacy during prolonged treatment in *in vitro* and animal models of biofilm- and non-biofilm-related *S. epidermidis* infections, and their effect on long-term population clearance and development of resistance and persistence.

Data and methodology

The methods are detailed and there is also detailed documentation of the results.

However, the way the description and results are presented makes the manuscript quite difficult to digest.

RESPONSE 3.3: Thank you for this comment. We have made two changes to make it easier for the reader to get an overview of the results: 1) We have divided the main text into a section headed "Introduction" and a section headed "Results and discussion"; 2) We have summarised our findings related to the 138 MRSE isolates in a new table (see Table 1),

Suggested improvements and clarity and context

There are a lot of results and figures generated based on the previous collection of strains (from reference 4). The manuscript might be easier to read by just describing the strains that became susceptible to 'amx/clavulanate' to demonstrate and highlight the key findings. The manuscript could be shortened and more succinct. The remainder/ complete set of results could just go to supplementary files. While the whole collection of strains is delineated into genotypes/clades, it may not be very relevant except that the effect is observed from strains isolated from laboratories in several countries.

RESPONSE 3.4: Thank you for these suggestions. Although we prefer to keep all the results in the main text, we would of course be happy to shorten the manuscript if the editor requests it. We would also like to keep the lineage/sublineage designations (i.e., BPH0662, ST-mixed, ST5, and ST23) as they are quite distinct in terms of resistance to various antibiotics, including amoxicillin/clavulanic acid and the last-resort antibiotics vancomycin and rifampicin, as well as of their ability to form biofilms.

REVIEWERS' COMMENTS

Reviewer #1 (Remarks to the Author):

The authors have mostly dealt with my concerns, and have added appropriate limitations to the text.

Reviewer #3 (Remarks to the Author):

The manuscript's clarity has improved, and the noteworthy results of susceptibility of MRSE to betalactam/betalactamase inhibitor in a proportion of *mecA*-positive isolates are more clearly spelled out. These strains are more likely falling into the category of ST23 and ST5 (and a few ST2 mixed) and have poor biofilm-forming ability in vitro. However, the experiments also demonstrate that these isolates develop phenotypic resistance in vitro with time, and remain a major limitation of using this combination for treatment.

The abstract could be rewritten to highlight the above results. Line 29-30 should be rephrased or removed, as a single-dose beta-lactam/beta-lactamase inhibitor in the mouse thigh model led to a log reduction in bacterial colony count, but would not be considered 'effective' as a form of treatment. The last sentence of the abstract should also be removed. It would be too preliminary to support its clinical value, especially with its rapid in vitro resistance development. There is also no evidence in the manuscript to support adding in vancomycin with the beta-lactam/beta-lactamase inhibitor to potentiate treatment.

Note the error in the legend in both Figs 1 and 2, which should read *Staph epidermidis* NOT *Staph aureus*.

Reviewer #1 (Remarks to the Author):

The authors have mostly dealt with my concerns, and have added appropriate limitations to the text.

Reviewer #3 (Remarks to the Author):

The manuscript's clarity has improved, and the noteworthy results of susceptibility of MRSE to betalactam/betalactamase inhibitor in a proportion of mecA-positive isolates are more clearly spelled out. These strains are more likely falling into the category of ST23 and ST5 (and a few ST2 mixed) and have poor biofilm-forming ability in vitro. However, the experiments also demonstrate that these isolates develop phenotypic resistance in vitro with time, and remain a major limitation of using this combination for treatment.

The abstract could be rewritten to highlight the above results. Line 29-30 should be rephrased or removed, as a single-dose beta-lactam/beta-lactamase inhibitor in the mouse thigh model led to a log reduction in bacterial colony count, but would not be considered 'effective' as a form of treatment. The last sentence of the abstract should also be removed. It would be too preliminary to support its clinical value, especially with its rapid in vitro resistance development. There is also no evidence in the manuscript to support adding in vancomycin with the beta-lactam/beta-lactamase inhibitor to potentiate treatment.

RESPONSE 3.1: In the revised Abstract, we have rephrased the sentence in l. 29-30, removed the last sentence, and emphasised the need to further address the in vivo risk of resistance development.

Note the error in the legend in both Figs 1 and 2, which should read Staph epidermidis NOT Staph aureus.

RESPONSE 3.2: We have changed "aureus" to "epidermidis" in both figure legends.